# Clinical Outcome and Prognosis of a Nosocomial Outbreak of COVID-19

**DOI:** 10.3390/jcm12062279

**Published:** 2023-03-15

**Authors:** Sang Hyuk Kim, Taehee Kim, Hayoung Choi, Tae Rim Shin, Yun Su Sim

**Affiliations:** 1Division of Pulmonary, Allergy, and Critical Care Medicine, Department of Internal Medicine, Hallym University Kangnam Sacred Heart Hospital, Seoul 07441, Republic of Korea; 2Lung Research Institute, Hallym University College of Medicine, Chuncheon 24252, Republic of Korea

**Keywords:** COVID-19, nosocomial infection, hospital-acquired infection

## Abstract

Nosocomial coronavirus disease 2019 (COVID-19) outbreaks have been reported despite widespread quarantine methods to prevent COVID-19 in society and hospitals. Our study was performed to investigate the clinical outcome and prognosis of a nosocomial outbreak of COVID-19. We retrospectively analyzed the medical records of patients diagnosed with nosocomial COVID-19 of severe acute respiratory syndrome coronavirus-2 (SARS-CoV-2) at a university teaching hospital between 1 November 2021 and 31 April 2022. Nosocomial COVID-19 was defined as a positive SARS-CoV-2 polymerase chain reaction (PCR) test result 4 or more days after admission in asymptomatic patients who had a negative SARS-CoV-2 PCR test on admission. In this study, 167 patients were diagnosed with nosocomial COVID-19 (1.14%) among a total of 14,667 patients admitted to hospital during the study period. A total of 153 patients (91.6%) survived, but 14 patients (8.4%) died. The median time between admission and COVID-19 diagnosis was 11 days, and the median duration of hospital stay was 24 days. After adjusting for other factors, no vaccination (adjusted HR = 5.944, 95% CI = 1.626–21.733, *p* = 0.007) and chronic kidney disease (adjusted HR = 6.963, 95% CI = 1.182–41.014, *p* = 0.032) were found to increase mortality risk. Despite strict quarantine, a significant number of nosocomial COVID-19 cases with a relatively high mortality rate were reported. As unvaccinated status or chronic kidney disease were associated with poor outcomes of nosocomial COVID-19, more active preventive strategies and treatments for patients with these risk factors are needed.

## 1. Introduction

Coronavirus disease 2019 (COVID-19) is still causing extensive damage worldwide despite the development of treatment methods and vaccines [1]. As COVID-19 is highly contagious, has a long incubation period, and is airborne, and considering the burden of in-hospital COVID-19 outbreaks, hospitals have established numerous prevention strategies, including providing caregivers with personal protective equipment (PPE), limiting visiting hours for general wards and intensive care units (ICUs), and frequent testing of patients and their caregivers for COVID-19 [2,3,4]. Nevertheless, nosocomial COVID-19 outbreaks have been reported continuously during the pandemic [5,6,7,8,9,10,11,12,13,14,15,16,17,18].

Like other viruses, COVID-19 has mutated over time [19]. The Omicron variant of COVID-19 was first reported on 9 November 2021 and has since become the dominant strain worldwide [20]. In Korea, the Omicron variant was the dominant strain from late 2021 to early 2022, peaking in March 2022 [21,22]. The Omicron variant of COVID-19 has an increased affinity for angiotensin-converting enzyme 2 (ACE2) receptors compared to previous variants (Alpha and Delta), resulting in high transmissibility [20]. This led to an increased risk of COVID-19 outbreaks among hospitalized patients [14,16,17,18]. In Korea, a nosocomial infection outbreak by Omicron was reported in a rehabilitation ward of a single hospital in January 2022 [16].

Previous studies reported that SARS-CoV-2 infections in hospitals accounted for 1.67–16.4% of all COVID-19 cases [6,7,12,23,24]. The incidence of nosocomial COVID-19 caused by Omicron variant infection has been reported to be 4.7% [25]. Among patients screened, mainly for index case patients in hospitals, there were also studies in which the incidence of nosocomial COVID-19 patients ranged from 15.5 to 30.9% [14,17]. In addition, hospitalized patients with nosocomial COVID-19 have increased severe outcomes, such as prolonged hospitalization and death [6,23]. However, one study that showed the mortality rate of nosocomial COVID-19 patients was lower due to having timely support compared to community-acquired COVID-19 patients admitted to hospital [6]. In a study that analyzed 200 patients with nosocomial COVID-19 by the Omicron variant, the patient incidence rate and clinical aspects were analyzed using COVID-19 screening of index patients [17]. There have been studies that have analyzed the incidence and prognosis of nosocomial COVID-19 caused by Omicron in healthcare workers and caregivers [16,18].

This study aims to evaluate the epidemiology of COVID-19 infection in hospital and risk factors for mortality by analyzing the incidence, clinical condition, and prognosis of nosocomial COVID-19 in hospitalized patients under quarantine during the epidemic period of the Omicron variant. 

## 2. Materials and Methods

### 2.1. Study Design

This study was conducted at a university teaching hospital in Seoul, Korea, between 1 November 2021 and 31 April 2022. The hospital has 578 beds (495 beds in general wards and 83 beds in the ICU) and 24 departments of subspecialties. We retrospectively reviewed the medical records of patients diagnosed with nosocomial COVID-19 due to SARS-CoV-2 infections.

All patients could only be admitted if the SARS-CoV-2 polymerase chain reaction (PCR) test was negative within 48 h before admission. Nosocomial COVID-19 was defined as a positive result of SARS-CoV-2 PCR after 4 days from admission in patients who were asymptomatic and showed a negative SARS-CoV-2 PCR upon admission. SARS-CoV-2 PCR tests were performed on patients in the hospital who showed symptoms. Furthermore, COVID-19 PCRs were performed on patients on the same days, 4 and 7 days after a positive diagnosis of COVID-19 in another patient or caregiver in the same room.

All specimens were obtained from the upper respiratory tract using a nasopharyngeal or throat swab. The SARS-CoV-2 PCR tests were performed using Standard M nCoV Real-Time Detection Kits (SD Biosensor, Suwon, Korea).

### 2.2. Hospital Quarantine against COVID-19

For COVID-19 patients, the hospital has 12 isolated beds in the intensive care unit (ICU), and 8 isolated beds in general wards. If a patient was diagnosed with nosocomial COVID-19, they were moved to COVID-19 unit and treated. All patients admitted to the ward must have shown no respiratory infection symptoms and a negative SARS-CoV-2 PCR within 48 h before admission. Patients who had already contracted COVID-19 could be hospitalized without additional COVID-19 PCRs for up to 45 days after confirming infection.

The number of caregivers for a patient was limited to one, and SARS-CoV-2 PCR tests had to be negative within 48 h of starting care. Visits to patients admitted to the general ward were prohibited. Family visits to patients admitted to the ICU were limited to less than 30 min once a day, and all visitors had to have a negative COVID-19 PCR within 3 days or complete vaccination records.

Only one entrance was used in each of the three buildings with inpatient wards. Persons with respiratory symptoms were prohibited from entering the hospital, and patients visiting outpatient clinics could visit after checking for fever and entering their contact information.

All patients and guardians were required to wear a face mask and medical staff were required to wear masks with a Korean filter (KF) of 94 or higher, capable of filtering out more than 94% of fine particles with an average size of 0.6μm. When treating patients with suspected COVID-19, disposable gloves, disposable long-sleeved gowns, disposable face shields, and masks with KF94 or higher were used.

### 2.3. Variables

The demographics of the patients, including age, sex, body mass index (BMI), and comorbidities, were recorded. We evaluated clinical records upon admission, including the department, diagnosis, vaccination status, and type of hospital room. Mortality was defined as the number of patients who died during hospitalization. Records of deaths were reviewed by the authors, and a consensus was reached as to whether they were related to nosocomial COVID-19. Old cerebrovascular disease was defined as brain hemorrhage or old cerebral infarction. Chronic airway disease was defined as chronic obstructive pulmonary disease and asthma. Chronic heart disease was defined as heart failure and old myocardial infarction. Chronic kidney disease was defined as glomerular filtration rate (GFR) < 50. The National Early Warning Score (NEWS) [26] was evaluated at the time of diagnosis of nosocomial COVID-19. NEWS was calculated using seven items—heart rate, temperature, mental status, systolic blood pressure, any supplemental oxygen, oxygen saturation, and respiration rate—that were scored on a scale from 0 to 20 points [26]. We also investigated symptoms, status before nosocomial COVID-19, and medication for COVID-19. Mortality was assessed as death during hospitalization in patients with COVID-19.

### 2.4. Statistical Analyses

Descriptive data are presented as the median with interquartile range (IQR) values, and frequencies are expressed as a number (%). Continuous variables were compared using the Mann–Whitney U test, and categorical variables were compared using Fisher’s exact test. Univariate analysis of risk factors for mortality was performed by creating a categorized variable based on the median values of laboratory and clinical characteristics in patients with COVID-19. Factors related to survival were analyzed in a Cox proportional hazard model to modify for the potential confounding effects of each factor. The results are shown as the adjusted hazard ratio (HR) with 95% confidence interval (CI). The 14 variables included sex, age, and variables associated with *p* < 0.25 in univariate analysis in Cox’s regression analysis. Cumulative survival curves were derived using the Cox proportional hazard model with vaccination status and chronic kidney disease. Statistical analysis was performed using SPSS software (version 18). In all analyses, *p* < 0.05 was taken to indicate statistical significance.

### 2.5. Ethics Statement

This study was conducted in accordance with the Declaration of Helsinki. Patient information was de-identified and anonymized before analysis, and informed consent was waived. This study was approved by the institutional review board of Hallym University Kangnam Sacred Heart Hospital (HKS 2019-12-016-002).

## 3. Results

From 14,667 admitted patients, 167 patients were diagnosed with nosocomial COVID-19 (1.14%). Of these, 153 patients (91.6%) survived, but 14 patients (8.4%) died. The incidence of patients with COVID-19 was lower in the hospital than in the whole country (Figure 1A).

### 3.1. Clinical Characteristics

The clinical characteristics of patients with nosocomial COVID-19 are shown in Table 1. The proportion of males was 63%, and the median age and BMI were 69 years and 23.4 kg/m^2^, respectively. Age was higher in non-survivors than in survivors (*p* = 0.047). There were no differences in sex or BMI between non-survivors and survivors.

The most common comorbidity was hypertension (50.0%), followed by diabetes (32.0%) and cancer (25%). The department with the highest incidence rate for nosocomial COVID-19 was orthopedics (19%), followed by pulmonology (17%) and gastroenterology (14%). The most common diagnosis was cancer (19.0%), followed by fractures (18.0%) and cerebrovascular accidents (CVAs, 18%). The proportion of patients diagnosed with pneumonia in non-survivors was higher than in survivors (*p* = 0.039).

With regard to vaccination status, 130 (78%) of the patients who developed COVID-19 in the hospital were vaccinated, and approximately two-thirds (88/130, 67.7%) of these patients had received three vaccination doses. Vaccination rates were significantly different between survivors and non-survivors. In terms of COVID-19 vaccinations, tozinameran was the most common first dose, and the number of patients who received tozinameran in the third dose was the highest (85.1%). Among patients who had never been vaccinated, 64.3% died compared to 18.3% in the surviving group (*p* < 0.001). The most common type of hospital room was a six-person room (52%).

### 3.2. Hospital Course and Outcomes

As shown in Table 2, the median time between admission to COVID-19 diagnosis was 11 days, and the median length of hospital days was 24 days. These durations were similar for non-survivors and survivors. Patients with COVID-19 stayed in the hospital longer than other patients, and the length of hospital stay was the longest in the department of infectious diseases (Figure 1B). Nearly half of the patients were asymptomatic (52%), and the most common symptom was fever (28%). Survivors were more likely to be asymptomatic than non-survivors (54% vs. 21%, *p* = 0.024). The rate of dyspnea was significantly higher in non-survivors than in survivors (36% vs. 9.2%, *p* = 0.012).

There were no differences in the rates of ventilator, steroid, or antibiotic use between survivors and non-survivors. After a COVID-19 diagnosis, non-survivors showed higher rates of pneumonia (71% vs. 11%, *p* < 0.001), sepsis (64% vs. 1.3%, *p* < 0.001), ventilator use (43% vs. 2.6%, *p* < 0.001), and change of antibiotics (79% vs. 19%, *p* < 0.001) than the survivors. The median NEWS was 2 points and was higher in non-survivors than in survivors (4 (range: 3–6) vs. 2 (range: 1–3) points, *p* < 0.001). The main medication administered for the treatment of COVID-19 was remdesivir (87%), followed by Paxlovid (1.8%). There were no differences in the rates of administration of these drugs between non-survivors and survivors.

### 3.3. Factors Associated with Mortality

Table 3 shows which factors were associated with mortality in patients with nosocomial COVID-19. In the unadjusted model, no vaccination (HR = 6.747, 95% CI = 2.206–20.145, *p* = 0.001), pneumonia (HR = 3.621, 95% CI = 1.135–11.547, *p* = 0.030), NEWS ≥ 2 (HR = 8.538, 95% CI = 1.117–65.273, *p* = 0.039), and steroid before diagnosis (HR = 3.905, 95% CI = 1.089–14.003, *p* = 0.037) showed a significantly increased risk of mortality. No vaccination (adjusted HR = 5.944, 95% CI = 1.626–21.733, *p* = 0.007) and chronic kidney disease (adjusted HR = 6.963, 95% CI = 1.182–41.014, *p* = 0.032) increased the risk of mortality after adjustment. Figure 2 shows survival curves according to no vaccination and chronic kidney disease from the time of hospitalization and the time of diagnosis of nosocomial COVID-19.

## 4. Discussion

This study showed that 1.14% of patients developed COVID-19 during hospitalization after the emergence of the Omicron variant. The incidence of nosocomial COVID-19 before the predominance of Omicron variant COVID-19 was reported to vary from 1.67 to 16.4% [6,7,12,23,24]. During the Omicron winter surge from 2021 to 2022, one study reported that the incidence of nosocomial COVID-19 was 4.7%, with 178 cases diagnosed out of 3820 [25]. Among patients mainly screened for index case patients in hospitals, some studies also reported that the incidence of nosocomial COVID-19 patients ranged from 15.5 to 30.9% [14,17]. The incidence of this nosocomial COVID-19 is inevitably influenced by the contagious power of this COVID-19 variant, the quarantine system of society and hospitals, and the systemic and immune status of hospitalized patients. Korea has maintained a firm quarantine policy since the beginning of the COVID-19 pandemic [27]. As mentioned above, our hospital also implemented strict policies to contain the coronavirus outbreak, including frequent COVID-19 tests for admitted patients, limiting the number of caregivers, and prohibition of visitors. Therefore, the incidence of nosocomial COVID-19 in this study may have been slightly lower than in other studies for Omicron nosocomial COVID-19, but the incidence of nosocomial COVID-19 increased to 4.36% when the Omicron variant surged the most in March 2022. In addition, it is possible that the incidence of nosocomial COVID-19 in this study was underestimated, as asymptomatic nosocomial COVID-19 cases accounted for 52% of the total cases. As we screened for the occurrence of nosocomial COVID-19, focusing on symptomatic patients or contacts of COVID-19 patients, asymptomatic patients without a history of contact with COVID-19 patients may not have been diagnosed with nosocomial COVID-19.

The mortality rate of nosocomial COVID-19 was 8.4%, which was significantly higher than the mortality rate of 0.12% for COVID-19 patients nationwide during the same period [9]. Previous studies have reported mortality rates ranging from 27.1 to 57.1% in patients with nosocomial COVID-19 [5,6,9,12,24]. In the nosocomial COVID-19 study with the predominance of the Omicron variant, the mortality rate ranged from 0 to 4.5% [14,16,17]. The mortality rate of nosocomial COVID-19 patients in our study was lower than that reported in the nosocomial COVID-19 study before the outbreak of the Omicron variant but higher than in other nosocomial COVID-19 studies during the Omicron dominant period. It was found that the fatality rate of the Omicron variant of COVID-19 is low compared to other variants [28]. However, the mortality rate of nosocomial infection is influenced by various aspects, such as the general condition, comorbidities, immune status, and severity of diagnosis in patients [29]. In a study of Omicron nosocomial COVID-19, patients in the head and neck surgery ward had a mortality rate of zero [14], and another study found a mortality rate of 1.1% for rehabilitation medicine ward patients, healthcare workers, and caregivers [16]. On the other hand, the mortality rate of nosocomial COVID-19 in patients with hematologic malignancy was very high (57.1%) before the outbreak of the Omicron variant [5]. Therefore, it is judged that the mortality of nosocomial COVID-19 patients in this study was influenced not only by the lethality of the Omicron variant but also by the severity of the disease, systemic and immune status, and comorbidities of the patient group in various aspects.

The NEWS was significantly higher in the non-survivors than the survivors in our study population. Nevertheless, the NEWS did not exceed the cut-off (NEWS ≥ 7) for rapid response system activation in most of the non-survivors [30]. Therefore, in addition to the conventional NEWS, efforts to detect high-risk groups for nosocomial COVID-19-associated mortality and intervention at an early stage are required.

In our study, the length of hospital stay in patients with nosocomial COVID-19 in each department was 2.9–6.6-fold longer than the average length of stay in each department during the same period. Several types of nosocomial infections are known to incur additional medical costs and an increased length of stay [31]. Hospitalization for approximately 2 weeks was required after diagnosis with nosocomial COVID-19 in our study population. Therefore, effective preventive measures are required to prevent damage caused to the medical system by nosocomial COVID-19.

Among the factors related to nosocomial COVID-19-associated mortality, unvaccinated status and chronic kidney disease had significant impacts. Although the vaccination rate for COVID-19 in Korea is high, there is still a proportion of the population who are unvaccinated [32]. In particular, some patients with underlying diseases may not be vaccinated due to concerns about side effects [33,34]. As these patients frequently visit hospitals, they are at a much higher risk of exposure to COVID-19 and nosocomial infection [35]. In addition, vaccinated individuals have been shown to have a decreased risk of COVID-19-associated mortality [36]. Therefore, public health authorities should devise a means of vaccinating individuals with underlying diseases. In this study, it was found that nosocomial COVID-19 patients with chronic kidney disease had a high risk of mortality. Another nosocomial COVID-19 study conducted in April 2020 also reported a high risk of death in patients with reduced renal function [6]. The impairment of normal reactions of the innate and adaptive immune systems in chronic kidney disease predisposes patients to an increased risk of infections, diminishing vaccine responses [37]. Therefore, in patients with chronic kidney disease, careful consideration should be given to pre-emptive protective measures against nosocomial COVID-19 and vaccination status at the time of hospitalization.

Our study has several limitations. First, the precise transmission pathway of COVID-19 is uncertain. We could not analyze the manner in which COVID-19 spread because there was no information regarding the route of infections, such as real-time genome sequencing [38]. Fortunately, protocols for hospital-acquired COVID-19 diagnosis have been published in the UK [39]. Future studies based on these protocols may be able to suggest solutions for problems related to nosocomial COVID-19, including information on the route of infections. Secondly, this study was conducted in a single teaching hospital in Korea that treats a significant number of patients with severe conditions. Thirdly, our study had a retrospective study design and a small sample size. However, we tried to include variables considering various clinical situations in the analysis. Fourthly, mortality was assessed only during hospitalization. If prognosis had been investigated using various methods, such as intensive care unit mortality or post-discharge mortality, more information on the clinical course of patients with nosocomial COVID-19 would have been provided. Additionally, because there were many asymptomatic nosocomial COVID-19 patients in this study, it is possible that the real incidence of nosocomial COVID-19 infection was underestimated. To address these limitations, it is hoped that a prospective cohort study that regularly conducts COVID-19 PCRs will be conducted.

In conclusion, despite strict quarantine protocols, a significant number of nosocomial COVID-19 cases occurred, with a relatively high mortality rate. As unvaccinated status and chronic kidney disease are associated with poor outcomes for nosocomial COVID-19, more active preventive strategies and treatments for patients with these risk factors are needed.

## Figures and Tables

**Figure 1 jcm-12-02279-f001:**
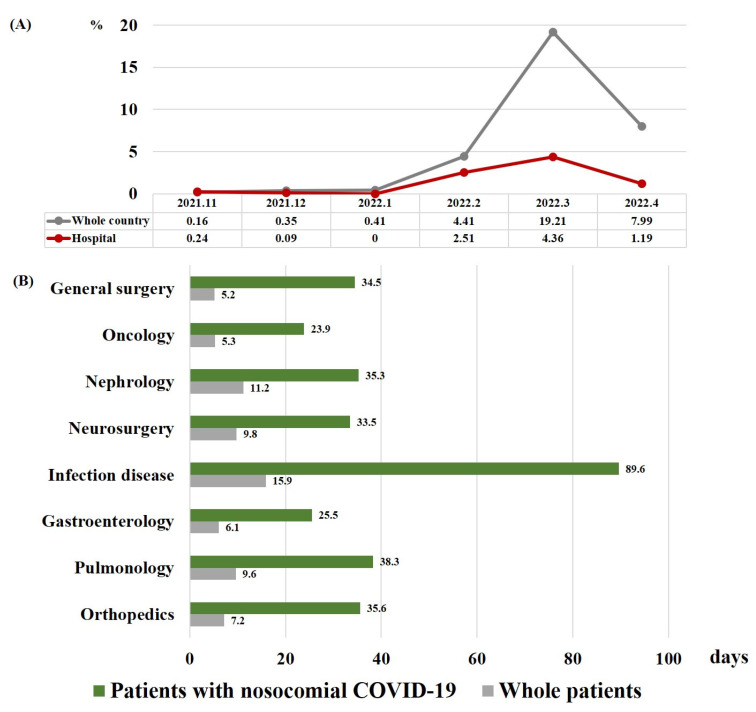
(**A**) Incidence of nosocomial COVID-19 in all hospitalized patients and of COVID-19 in the population of the country as a whole. (**B**) Median length of hospital stay for patients with nosocomial COVID-19 and for all patients in each department. Abbreviation: COVID-19, coronavirus disease 2019.

**Figure 2 jcm-12-02279-f002:**
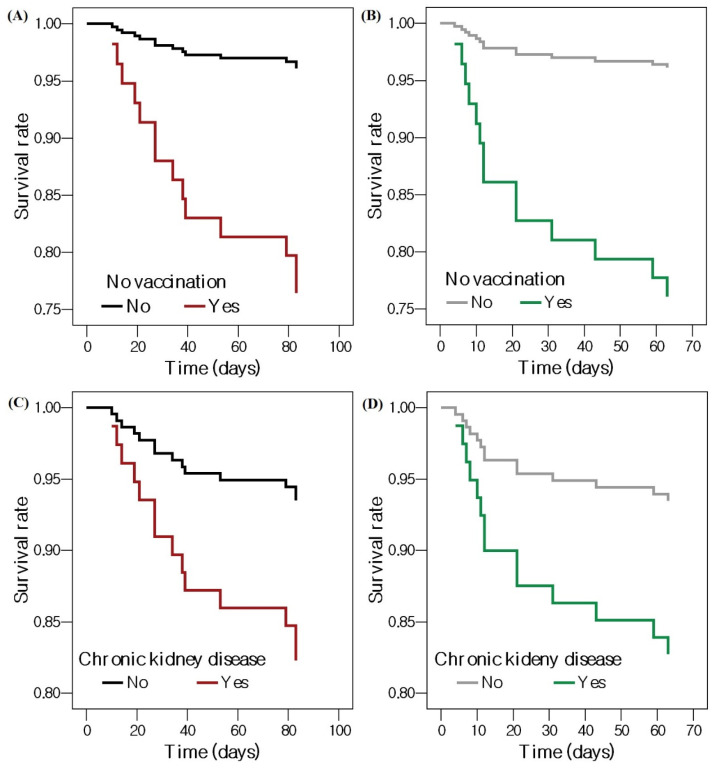
Cox proportional survival curves showing the effects of no vaccination on patients with nosocomial COVID-19 (**A**) from the time of admission and (**B**) from the time of diagnosis with nosocomial COVID-19, and chronic kidney disease in patients with nosocomial COVID-19 (**C**) from the time of admission and (**D**) from the time of diagnosis with nosocomial COVID-19. Abbreviation: COVID-19, coronavirus disease 2019.

**Table 1 jcm-12-02279-t001:** Clinical characteristics of patients with nosocomial COVID-19 (n = 167).

Variable	All Patients (n = 167)	Survivor (n = 153)	Non-Survivor (n = 14)	*p*-Value
Sex, male	105 (63%)	97 (63%)	8 (57%)	0.774
Age, year	69 (57–81)	68 (57–79)	78 (66–89)	0.047
BMI, kg/m^2^	23.4 (20.8–26.2)	23.4 (20.1–26.2)	21.9 (18.6–24.5)	0.126
Co-morbidity				
Hypertension	84 (50%)	76 (50%)	8 (57%)	0.781
Diabetes	53 (32%)	47 (31%)	6 (43%)	0.376
Cancer	42 (25%)	36 (24%)	6 (43%)	0.119
Chronic kidney disease	29 (17%)	24 (16%)	5 (36%)	0.071
History of CVA	24 (14%)	20 (13%)	4 (28.6%)	0.121
Chronic airway disease	13 (7.8%)	10 (6.5%)	3 (21%)	0.081
Liver cirrhosis	11 (6.6%)	9 (5.9%)	2 (14.3%)	0.232
Congestive heart failure	10 (6.0%)	10 (6.5%)	0	1.000
Department				
Orthopedics	32 (19%)	32 (21%)	0	0.074
Pulmonology	28 (17%)	23 (15%)	5 (36%)	0.062
Gastroenterology	23 (14%)	20 (13%)	3 (21%)	0.414
Infection disease	18 (11%)	17 (11%)	1 (7%)	1.000
Neurosurgery	15 (9.0%)	15 (9.2%)	0	0.618
Nephrology	15 (9.0%)	13 (8.5%)	2 (14.3%)	0.365
Oncology	12 (7.2%)	10 (6.5%)	2 (14.3%)	0.265
General surgery	6 (3.6%)	6 (3.9%)	0	1.000
Cardiology	5 (3.0%)	5 (3.3%)	0	1.000
Neurology	4 (2.3%)	4 (2.6%)	0	1.000
Others	9 (5.4%)	8 (5.2%)	1 (7.1%)	0.554
Diagnosis				
Cancer	31 (19%)	28 (19%)	3 (21%)	0.726
Fracture	18 (11%)	17 (11%)	1 (7%)	1.000
CVA	18 (11%)	18 (12%)	0	0.368
Pneumonia	17 (10%)	13 (8.5%)	4 (29%)	0.039
Abdomen infection	16 (9.6%)	16 (11%)	0	0.366
Osteomyelitis or arthritis	16 (9.6%)	14 (9.2%)	2 (14%)	0.627
Spine fracture and stenosis	11 (6.6%)	11 (7.2%)	0	0.602
Urinary tract infection	10 (6.0%)	9 (5.9%)	1 (7.1%)	0.594
Acute or chronic renal failure	9 (5.4%)	9 (5.9%)	0	1.000
Others	21 (12%)	18 (11.8%)	3 (21.4%)	0.389
Vaccination				
None	37 (22%)	28 (18.3%)	9 (64.3%)	<0.001
First	7 (4.2%)	7 (4.6%)	0	1.000
Tozinameran	5 (71.4%)			
Covishield	2 (28.6%)			
Second	36 (21.6%)	36 (23.5%)	0	0.042
Tozinameran	20 (55.6%)			
Covishield	10 (27.8%)			
Elasomeran	6 (16.7%)			
Third	87 (52.1%)	82 (53.6%)	5 (35.7%)	0.266
Tozinameran	74 (85.1%)	71 (86.6%)	3 (60%)	
Elasomeran	13 (14.9%)	11 (13.4%)	2 (40%)	
Type of hospital room				
Single person room	14 (8.4%)	11 (7.2%)	3 (21.4%)	0.098
Three-person room	9 (5.4%)	9 (5.9%)	0	1.000
Four-person room	46 (28%)	42 (28%)	4 (29%)	1.000
Five-person room	8 (4.8%)	6 (3.9%)	2 (14.3%)	0.137
Six-person room	87 (52%)	82 (53.6%)	5 (35.7%)	0.266
Intensive care unit	3 (1.8%)	3 (2.0%)	0	1.000

Data for categorical and continuous variables are presented as numbers (%) and median (interquartile ranges). COVID-19, coronavirus disease 2019; BMI, body mass index; CVA, cerebrovascular disease.

**Table 2 jcm-12-02279-t002:** Hospital course and outcomes of patients with nosocomial COVID-19.

Variable	All patients (n = 167)	Survivor (n = 153)	Non-Survivor (n = 14)	*p*-Value
Duration, days				
From admission to COVID-19 infection	11 (7–23)	10 (7–23)	12 (8–34)	0.344
Hospital days	24 (15–42)	24 (15–43)	27 (14–43)	0.995
Symptoms				
Fever	47 (28%)	40 (26%)	7 (50%)	0.068
Cough	28 (17%)	25 (16%)	3 (21%)	0.707
Dyspnea	9 (11%)	14 (9.2%)	5 (36%)	0.012
Asymptomatic state	86 (52%)	83 (54%)	3 (21%)	0.024
Status before COVID-19				
Ventilator	4 (2.4%)	3 (2.0%)	1 (7.1%)	0.298
Antibiotics	101 (61%)	90 (59%)	11 (79%)	0.168
Steroid	11 (6.6%)	8 (5.2%)	3 (21%)	0.052
Status after COVID-19				
Pneumonia	27 (16%)	17 (11%)	10 (71%)	<0.001
Sepsis	11 (6.6%)	2 (1.3%)	9 (64%)	<0.001
Ventilator	10 (6.0%)	4 (2.6%)	6 (43%)	<0.001
Change of antibiotics	40 (24%)	29 (19%)	11 (79%)	<0.001
NEWS	2 (1–4)	2 (1–3)	4 (3–6)	<0.001
Medication for COVID-19				
Remdesivir	145 (87%)	131 (86%)	14 (100%)	0.219
Paxlovid	3 (1.8%)	3 (2.0%)	0	1.000

Data for categorical and continuous variables are presented as numbers (%) and medians (interquartile ranges). COVID-19, coronavirus disease 2019; NEWS, National Early Warning Score.

**Table 3 jcm-12-02279-t003:** Factors associated with mortality in patients with nosocomial COVID-19.

Characteristics	Unadjusted HR	95% CI	*p*-Value	Adjusted HR	95% CI	*p*-Value
Age ≥ 70 years	1.335	0.463–3.847	0.593	4.909	0.853–28.240	0.075
Male	0.775	0.269–2.235	0.638	0.943	0.264–3.375	0.928
BMI ≤ 23.4 kg/m^2^	1.368	0.475–3.943	0.562			
No vaccination	6.747	2.260–20.145	0.001	5.944	1.626–21.733	0.007
Comorbidity						
Cancer	2.392	0.930–6.896	0.106	2.713	0.687–10.713	0.154
Chronic airway disease	3.409	0.950–12.230	0.060	2.088	0.295–14.781	0.461
Chronic kidney disease	2.901	0.972–8.658	0.056	6.963	1.182–41.014	0.032
History of CVA	2.495	0.782–7.956	0.122	2.610	0.558–12.215	0.223
Liver cirrhosis	2.495	0.558–11.150	0.231	1.017	0.112–9.218	0.988
Single person room admission	3.104	0.856–11.133	0.082	3.030	0.587–15.642	0.186
Pulmonology	2.861	0.959–8.539	0.059	1.630	0.199–13.371	0.649
Pneumonia	3.621	1.135–11.547	0.030	4.908	0.669–36.028	0.118
NEWS ≥ 2	8.538	1.117–65.273	0.039	5.303	0.571–49.279	0.142
Antibiotics before diagnosis	2.434	0.679–8.724	0.172	2.354	0.405–13.663	0.340
Steroid before diagnosis	3.905	1.089–14.003	0.037	2498	0.479–13.018	0.277

COVID-19, coronavirus disease 2019; CVA, cerebrovascular disease; NEWS, National Early Warning Score; HR, hazard ratio; CI, confidence interval.

## Data Availability

The data presented in this study are available on request from the corresponding author.

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
