# Peer review of "Clinical Outcome and Prognosis of a Nosocomial Outbreak of COVID-19"

_jcm, 2023, doi:10.3390/jcm12062279_

Round 1

Reviewer 1 Report

Thank you for this manuscript entitled Clinical Outcome and Prognosis of a Nosocomial Outbreak of COVID-19. 

The subject is of interest, since publications regarding nosocomial COVID-19 are still scarce, and multiple evolution have occurred since the start of the pandemic (vaccination, variants, treatments). 

However, several major improvements need to be implemented in the manuscript.

Introduction:

The introduction must be improved with elements of context regarding nosocomial COVID in the Omicron era. Indeed, the study takes place in late 2021, early 2022, with Omicron probably being the major variant circulating in South Korea. Several reports have been published regarding nosocomial COVID-19 in the Omicron outbreaks, and in a mostly vaccinated population. 

The affirmation "prognosis of nosocomial COVID-19 and factors associated with several outcomes were poorly understood" is not correct. Actually, several recent reports (and older one) on healthcare acquired COVID-19 have been published, including an evaluation of outcomes and mortality in patients. I suggest reading the recently published study by Hawkins et al. in J Hosp Infect 2023 : Transmission dynamics and associated mortality of nosocomial COVID-19 throughout 2021 : a retrospective study at a large teaching hospital in London.

Going through the references cited in this paper could also help improve the introduction.

In Introduction or Methods, a description of SARS-CoV-2 variants circulating in South Korea at the period of the study would be needed.

Methods:

In methods, the description of screening practices in the setting is unclear. Were all patients screened upon admission ? Screening in the hospital or for programmed activities a test before admission had to be provided ? 

In paragraph 2.2, the first sentences are unclear. The reader understands that 12 ICUs and 8 general wards were dedicated to COVID-19 patients. That seems unlikely. If a patient had symptoms, or tested positive, where was he admitted ? Was there a "COVID-19" unit ? 

In methods, mortality was assessed only during hospitalisation. This is a limit that should be underlined in discussion. 

Results

Major improvements are needed in the Results section. All statistic tests should be done again, since they seem to be flawed. In 3.1 section, the authors state that "there were no differences in sex, age of BMI between survivors and non-survivors (p> 0.05 for all)". This statement is false, since in Table 1 age was significantly superior in non-survivors regarding p-value (0.047). I went through doing Fisher's exact test on several variables to control the authors' results, and I think all statistic tests must be done again. I found a systematic discrepancy, not major, but sometimes sufficient to alter the conclusions of the study. 

Vaccination status emerge as a variable of importance regarding survival of nosocomial COVID-19. However in methods, results or discussion, the authors provided no information regarding type of vaccines used in South Korea and used in the patients included in the study. 

In Table 2, the NEWS score should not be included in the Medication section of the table. Please move it to the "Status after COVID-19" section.

In Table 2, all statistics must be done again. For example in Fisher's exact test, steroid use before COVID-19 was not significant (p = 0.052). 

In Table 3, the authors use a cut-off at 69 years to assess factors associated with mortality. This cut-off was not explained anywhere and it is unclear why it was implemented. 

The symptoms are not relevant to this assessment. It seems obvious that asymptomatic state is not associated to mortality or dyspnea is associated to mortality. These variables must be taken off the statistics of this table and should not be discussed in discussion. All statistics should be done again, after removal of these variables. Figure 2 should be modified to remove Dyspnea from assessement. 

Discussion

The discussion will have to be written again regarding new statistics made and results obtained. 

The discussion was rather poor regarding comparison with other reports assessing nosocomial COVID-19 with Omicron variants and sub-variants. Discussing results regarding mortality from early in the pandemic does not seem relevant at this point. Several reports pointed out a 3 to 5% mortality for hospital-acquired COVID-19 with Omicron, the authors report a 8.2% mortality that seems superior and should be discussed (vaccination status ? comorbities ? availabilty of treatments ?). Only in-hospital mortality was assessed and this limit should be discussed. 

All discussion regarding dyspnea should be removed since irrelevant. 

I disagree with the authors about the number of patients included being large. This remains a small number study, moreover since the main assessments are made between survivors and non-survivors and there were only 14 non-survivors. This limits drastically the conclusions of this study.

There are some typos all along the manuscript, it should be read again and corrected.  

Author Response

Review 1

We sincerely appreciate the time and effort you invested to help us improve this manuscript.

We revised the manuscript (red color) and have provided a point-by-point response to your comments below.

Thank you for this manuscript entitled Clinical Outcome and Prognosis of a Nosocomial Outbreak of COVID-19.

The subject is of interest, since publications regarding nosocomial COVID-19 are still scarce, and multiple evolution have occurred since the start of the pandemic (vaccination, variants, treatments).

However, several major improvements need to be implemented in the manuscript.

Introduction:

The introduction must be improved with elements of context regarding nosocomial COVID in the Omicron era. Indeed, the study takes place in late 2021, early 2022, with Omicron probably being the major variant circulating in South Korea. Several reports have been published regarding nosocomial COVID-19 in the Omicron outbreaks, and in a mostly vaccinated population.

Thanks for pointing this important issue. As you pointed out, Omicron variant COVID 19 was prevalent in Korea at the time of this study, and hospital-acquired infections increased due to Omicron's high contagiousness. The introduction part has been rewritten by adding many reports on other nosocomial COVID 19 studies, especially the nosocomial COVID 19 studies by the Omicron variant, including the study of Hawkins you gratefully recommended.

The Omicron variant of severe acute respiratory syndrome coronavirus 2 (SARS-CoV-2) was first reported on November 9, 2021, and has been the dominant strain worldwide [20]. In Korea, the Omicron variant has been the dominant strain from late 2021 to early 2022, peaking in March 2022 [21,22].  (page3, line 30-33)

This leaded to an increased risk of COVID-19 outbreaks among hospitalized patients [14,16-18]. In Korea, a nosocomial infection outbreak by Omicron was reported in a rehabilitation ward of a single hospital in January 2022 [16]. (page3, line36-39)

Previous studies reported that SARS-CoV-2 infections in hospitals accounted for 1.67%–16.4% of all COVID-19 cases [6,7,12,23,24]. The incidence of nosocomial COVID 19 caused by Omicron variant infection has been reported to range from 4.7% [25]. Among patients screened mainly for index case patients in hospitals, there were also studies in which the incidence of nosocomial COVID 19 patients ranged from 15.5 to 30.9% [14,17]. (page3, line40 ~ page4, line 44)

However, there was also a study that showed that the mortality rate of nosocomial COVID 19 patients was lower due to having timely supportive compared to community acquired COVID19 patients admitted to hospital [6]. In a study that analyzed 200 patients with nosocomial COVID 19 by Omicron variant, the patient incidence rate and clinical aspects were analyzed through COVID 19 screening related to index patients [17]. There have been studies that have analyzed the incidence and prognosis of nosocomial COVID 19 caused by Omicron, including healthcare workers and caregivers [16,18]. (page4, line 46-53)

  1. Carter, B.; Collins, J.T.; Barlow-Pay, F.; Rickard, F.; Bruce, E.; Verduri, A.; Quinn, T.J.; Mitchell, E.; Price, A.; Vilches-Moraga, A., et al. Nosocomial COVID-19 infection: examining the risk of mortality. The COPE-Nosocomial Study (COVID in Older PEople). J Hosp Infect 2020, 106, 376-384.
  2. Zhou, Q.; Gao, Y.; Wang, X.; Liu, R.; Du, P.; Wang, X.; Zhang, X.; Lu, S.; Wang, Z.; Shi, Q., et al. Nosocomial infections among patients with COVID-19, SARS and MERS: a rapid review and meta-analysis. Ann Transl Med 2020, 8, 629.
  3. Borges, V.; Isidro, J.; Macedo, F.; Neves, J.; Silva, L.; Paiva, M.; Barata, J.; Catarino, J.; Ciobanu, L.; Duarte, S., et al. Nosocomial Outbreak of SARS-CoV-2 in a "Non-COVID-19" Hospital Ward: Virus Genome Sequencing as a Key Tool to Understand Cryptic Transmission. Viruses 2021, 13.
  4. Boshier, F.A.T.; Venturini, C.; Stirrup, O.; Guerra-Assuncao, J.A.; Alcolea-Medina, A.; Becket, A.H.; Byott, M.; Charalampous, T.; Filipe, A.D.S.; Frampton, D., et al. The Alpha variant was not associated with excess nosocomial SARS-CoV-2 infection in a multi-centre UK hospital study. J Infect 2021, 83, 693-700.
  5. Paltansing, S.; Sikkema, R.S.; de Man, S.J.; Koopmans, M.P.G.; Oude Munnink, B.B.; de Man, P. Transmission of SARS-CoV-2 among healthcare workers and patients in a teaching hospital in the Netherlands confirmed by whole-genome sequencing. J Hosp Infect 2021, 110, 178-183.
  6. Ponsford, M.J.; Jefferies, R.; Davies, C.; Farewell, D.; Humphreys, I.R.; Jolles, S.; Fairbairn, S.; Lewis, K.; Menzies, D.; Benjamin, A., et al. Burden of nosocomial COVID-19 in Wales: results from a multicentre retrospective observational study of 2508 hospitalised adults. Thorax 2021, 76, 1246-1249.
  7. Zhang, M.; Xiao, J.; Deng, A.; Zhang, Y.; Zhuang, Y.; Hu, T.; Li, J.; Tu, H.; Li, B.; Zhou, Y., et al. Transmission Dynamics of an Outbreak of the COVID-19 Delta Variant B.1.617.2 - Guangdong Province, China, May-June 2021. China CDC Wkly 2021, 3, 584-586.
  8. Itoh, N.; Akazawa, N.; Ishikane, M.; Kawabata, T.; Kawamura, D.; Chikusa, T.; Kodama, E.N.; Ohmagari, N. Lessons learned from an outbreak of COVID-19 in the head and neck surgery ward of a Japanese cancer center during the sixth wave by Omicron. J Infect Chemother 2022, 28, 1610-1615.
  9. Jung, J.; Lee, J.; Park, H.; Lim, Y.J.; Kim, E.O.; Park, M.S.; Kim, S.H. Nosocomial Outbreak by Delta Variant From a Fully Vaccinated Patient. J Korean Med Sci 2022, 37, e133
  10. Sohn, Y.J.; Shin, P.J.; Oh, W.S.; Kim, E.; Kim, Y.; Kim, Y.K. Clinical Characteristics of Patients Who Contracted the SARS-CoV-2 Omicron Variant from an Outbreak in a Single Hospital. Yonsei Med J 2022, 63, 790-793.
  11. Hawkins, L.P.A.; Pallett, S.J.C.; Mazzella, A.; Anton-Vazquez, V.; Rosas, L.; Jawad, S.M.; Shakespeare, D.; Breathnach, A.S. Transmission dynamics and associated mortality of nosocomial COVID-19 throughout 2021: a retrospective study at a large teaching hospital in London. J Hosp Infect 2023, 133, 62-69.
  12. Wong, S.C.; Chan, V.W.; Yuen, L.L.; AuYeung, C.H.; Leung, J.O.; Li, C.K.; Kwok, M.O.; So, S.Y.; Chen, J.H.; Chiu, K.H., et al. Infection of healthcare workers despite a high vaccination rate during the fifth wave of COVID-19 due to Omicron variant in Hong Kong. Infect Prev Pract 2023, 5, 100261.
  13. Klompas, M.; Pandolfi, M.C.; Nisar, A.B.; Baker, M.A.; Rhee, C. Association of Omicron vs Wild-type SARS-CoV-2 Variants With Hospital-Onset SARS-CoV-2 Infections in a US Regional Hospital System. JAMA 2022, 328, 296-298.
  14. Leducq, V.; Couturier, J.; Granger, B.; Jolivet, S.; Morand-Joubert, L.; Robert, J.; Denis, M.; Salauze, B.; Goldstein, V.; Zafilaza, K., et al. Investigation of healthcare-associated COVID-19 in a large French hospital group by whole-genome sequencing. Microbiol Res 2022, 263.

The affirmation "prognosis of nosocomial COVID-19 and factors associated with several outcomes were poorly understood" is not correct. Actually, several recent reports (and older one) on healthcare acquired COVID-19 have been published, including an evaluation of outcomes and mortality in patients. I suggest reading the recently published study by Hawkins et al. in J Hosp Infect 2023: Transmission dynamics and associated mortality of nosocomial COVID-19 throughout 2021 : a retrospective study at a large teaching hospital in London.

Thanks for pointing this important issue. We removed "prognosis of nosocomial COVID-19 and factors associated with several outcomes were poorly understood" in introduction. In addition, the introduction was rewritten by adding references to other nosocomial COVID 19 listed in the answer above, including Hawkins' study you recommended.

Going through the references cited in this paper could also help improve the introduction.

Thank you for your kindly comment. The references cited in Hawkins’ paper were very helpful, and additional studies on nosocomial COVID 19 by omicron variants were also added as references.

In Introduction or Methods, a description of SARS-CoV-2 variants circulating in South Korea at the period of the study would be needed.

Thank you for your advice. We additionally described the prevalence and nosocomial infection of Omicron variant COVID 19 in Korea at the time of our study in the introduction.

In Korea, the Omicron variant has been the dominant strain from late 2021 to early 2022, peaking in March 2022 [21]. (page3, line 32-33)

In Korea, a nosocomial infection outbreak by Omicron was reported in a rehabilitation ward of a single hospital in January 2022 [16]. (page3, line 37-38)

Methods:

In methods, the description of screening practices in the setting is unclear. Were all patients screened upon admission? Screening in the hospital or for programmed activities a test before admission had to be provided?

Thanks for pointing this important issue. All patients who are admitted to our hospital can be hospitalized only when the SARS-CoV-2 PCR test is negative within two days before admission. This is not clearly described, so I added and rewrote about this point in the methods section.

“All patients who are admitted to the hospital can be hospitalized only if the SARS-CoV-2 (PCR) test is negative within 48 hours before admission.” (page5 line66-67)

During hospitalization, if a patient who complains of COVID 19 symptoms or patients or caregiver of same room is diagnosed in COVID 19, SARS-CoV-2 PCR test is performed after the same day, four and seven days later.

In paragraph 2.2, the first sentences are unclear. The reader understands that 12 ICUs and 8 general wards were dedicated to COVID-19 patients. That seems unlikely. If a patient had symptoms, or tested positive, where was he admitted? Was there a "COVID-19" unit?

I am very sorry that I mentioned it wrong. The hospital has isolated 12 beds in ICU, and isolated 8 beds in general wards for COVID-19 patients. If the patient is diagnosed with nosocomial COVID 19, they move to COVID 19 unit (12 beds in ICU and 8 beds in general wards) and are treated in the hospital. We add this point in methods section. (page5, line75-77)

In methods, mortality was assessed only during hospitalisation. This is a limit that should be underlined in discussion.

Thank you for your advice. In the discussion section, this point was added and described in the limitations of the study.

Fourth, mortality was assessed only during hospitalization. If the prognosis was investigated by various methods, such as intensive care unit mortality or post-discharge mortality, more information on the clinical course of patients with nosocomial COVID 19 would have been provided. (page22, line 296-299)

Results

Major improvements are needed in the Results section. All statistic tests should be done again, since they seem to be flawed. In 3.1 section, the authors state that "there were no differences in sex, age of BMI between survivors and non-survivors (p> 0.05 for all)". This statement is false, since in Table 1 age was significantly superior in non-survivors regarding p-value (0.047). I went through doing Fisher's exact test on several variables to control the authors' results, and I think all statistic tests must be done again. I found a systematic discrepancy, not major, but sometimes sufficient to alter the conclusions of the study.

Thank you for your advice and I’m very sorry for our mistake. We rewrote that “Age was higher in non-survivors than in survivors (p=0.047). There were no differences in sex or BMI between survivors and non-survivors” (page10, line150-151) in results. We re-ran all the statistics as you recommended and re-edited the tables. For analysis of categorical variables, Fisher's exact test was used. As suggested by another reviewer, statistical analysis was performed separately for department, diagnosis, vaccination, and hospital room variables according to subdivided variables. The changed and added contents while rewriting Table 1 were added to the result section. In Table 1, only the department part was extracted and shown below, and all other parts were modified through statistical analysis again.

The proportion of patients diagnosed with pneumonia in non-survivors was higher than in survivors (p=0.039). (page12, line161-162)

Vaccination status emerge as a variable of importance regarding survival of nosocomial COVID-19. However, in methods, results or discussion, the authors provided no information regarding type of vaccines used in South Korea and used in the patients included in the study.

Thank you for your advice. For each patient who received only the 1st vaccination, the 2nd vaccination, and the 3rd vaccination, the vaccinations administered at the time are listed in Table 1 and results section as product names.

For the COVID 19 vaccination, tozinameran was the most common in the first, and the number of patients who received tozinameran in the third was the highest at 85.1%. Among patients who had never been vaccinated, 64.3% of patients died, compared to 18.3% in the surviving group (p<0.001). (page12, line166-169)

In Table 2, the NEWS score should not be included in the Medication section of the table. Please move it to the "Status after COVID-19" section.

Thank you for your comment. We tried to put “NEWS” in the table as a single row, but the location seems ambiguous. We moved location to “Status after COVID19”.

In Table 2, all statistics must be done again. For example, in Fisher's exact test, steroid use before COVID-19 was not significant (p = 0.052). (Fisher’s test)

Thank you for your comment. We re-ran all the statistics as you recommended and re-edited the tables and results. For analysis of categorical variables, Fisher's exact test was used

In Table 3, the authors use a cut-off at 69 years to assess factors associated with mortality. This cut-off was not explained anywhere and it is unclear why it was implemented.

Thank you for your comment. The reason of cut-off at 69 years was because we categorized it as the median value of the continuous variables in our study. However, in order to facilitate the reader's readability and clinical significance, the analysis was re-categorized based on the age of 70 years. We rewrote that in table 3.

\

The symptoms are not relevant to this assessment. It seems obvious that asymptomatic state is not associated to mortality or dyspnea is associated to mortality. These variables must be taken off the statistics of this table and should not be discussed in discussion. All statistics should be done again, after removal of these variables. Figure 2 should be modified to remove Dyspnea from assessement.

Thank you for your advice. We excluded “dyspnea” from the variables to evaluate the risk factor of mortality, and also excluded dyspnea from discussion and figure 2. In addition, at the recommendation of another reviewer, Cox’s regression analysis of risk factors for mortality was performed by creating a categorized variable based on univariate analysis in patients with COVID-19 with p < 0.25.

Discussion

The discussion will have to be written again regarding new statistics made and results obtained.

Thank you for your comment. Based on the contents of the newly performed statistical analysis, both the results and discussion were modified.

This study showed that 1.14% of patients developed COVID-19 during hospitalization after the emergence of the omicron variant. The reported incidence of nosocomial COVID 19 before the predominance of Omicron variant COVID 19 was reported to vary from 1.67 to 16.4%[6,7,12,23,24]. During the Omicron winter surge from 2021 to 2022, there was a study reporting that the incidence of nosocomial COVID 19 was 4.7% with 178 cases diagnosed out of 3820[25]. Among patients screened mainly for index case patients in hospitals, there were also studies in which the incidence of nosocomial COVID 19 patients ranged from 15.5 to 30.9% [14,17]. The incidence of this nosocomial COVID 19 is inevitably influenced by the contagious power of the COVID 19 variant, the quarantine system of society and hospitals, and the systemic and immune status of hospitalized patients. Korea has maintained a firm quarantine policy since the beginning of the COVID-19 pandemic [28]. As mentioned above, our hospital also implemented strong policies to contain the coronavirus outbreak, including frequent COVID-19 tests for admitted patients, limiting the number of caregivers and prohibition of visitors. Therefore, the incidence of nosocomial COVID 19 in this study may have been slightly lower than in other studies for Omicron nosocomial COVID 19, but the incidence of nosocomial COVID 19 increased to 4.36% when the Omicron variant surged the most in March 2022. In addition, it is possible that the incidence of nosocomial COVID 19 in this study was underestimated, since asymptomatic nosocomial COVID 19 patients in this study accounted for 52%. Because we screened for the occurrence of nosocomial COVID 19, focusing on symptomatic patients or contacts of COVID 19 patients, asymptomatic patients without history of contact with COVID 19 patients may not be diagnosed with nosocomial COVID 19. (page19)

The discussion was rather poor regarding comparison with other reports assessing nosocomial COVID-19 with Omicron variants and sub-variants. Discussing results regarding mortality from early in the pandemic does not seem relevant at this point. Several reports pointed out a 3 to 5% mortality for hospital-acquired COVID-19 with Omicron, the authors report a 8.2% mortality that seems superior and should be discussed (vaccination status ? comorbities ? availabilty of treatments ?). Only in-hospital mortality was assessed and this limit should be discussed.

Thanks for pointing this important issue. The incidence and mortality of nosocomial COVID 19 before and after the outbreak of the Omicron variant are described again in the discussion section. In other Omicron nosocomial COVID 19 studies, direct comparison was difficult due to insufficient severity description, but low mortality rates were shown in studies involving otolaryngology wards, rehabilitation medicine wards, health care workers, and caregiver, which are judged to be of relatively low severity, and although omicron pre-epidemic studies, hematologic malignancy patients is very high mortality of nosocomial COVID 19. It was judged that the severity of the patient group, comorbidities, and immune status would all have an impact.

Previous studies have reported mortality rates ranging from 27.1 to 57.1% in patients with nosocomial COVID-19 [5,6,9,12,24]. In the nosocomial COVID 19 study with the predominance of the Omicron variant, the mortality rate ranged from 0 to 4.5% [14,16,17]. The mortality rate of nosocomial COVID 19 patients in our study is lower than that reported in the nosocomial COVID 19 study before the outbreak of the Omicron variant, but higher than in other nosocomial COVID 19 studies during the Omicron dominant period. It has been found that the fatality rate of the Omicron variant COVID 19 is weak compared to other variants [30]. However, the mortality rate of nosocomial infection is influenced in various aspects by the general condition, comorbidities, immune status, and severity of diagnosis in patients [31]. In the Omicron nosocomial COVID 19 study, the subject group of the study in which the mortality rate was zero was head and neck surgery ward patients [14], and the study with 1.1% was rehabilitation medicine ward patients, health care workers, and caregivers [16]. On the other hand, the mortality rate of nosocomial COVID 19 in patients of the hematologic malignancy was very high at 57.1% before the outbreak of the Omicron variant [5]. Therefore, it is judged that the mortality of nosocomial COVID 19 patients in this study was influenced not only by the fatality of the Omicron variant, but also by the severity of the disease, systemic and immune status, and comorbidities of the patient group from various aspects. (page20, line 242-258)

All discussion regarding dyspnea should be removed since irrelevant.

Thank you for your advice. We excluded variables for symptoms, including dyspnea, from the analysis, and included variables with p-values below 0.25 as recommended by other reviewers, revealing no vaccination and chronic kidney disease as risk factors for mortality. further described.

In this study, it was found that nosocomial COVID 19 patients with chronic kidney disease had a high risk of mortality. Another nosocomial COVID 19 study conducted in April 2020 also reported a high risk of death in patients with reduced renal function [6]. Impairment of the normal reaction of the innate and adaptive immune systems in chronic kidney disease predisposes patients to an increased risk of infections, diminished vaccine response [39]. Therefore, in patients with chronic kidney disease, careful consideration should be given to preemptive protective measures against nosocomial COVID 19 and vaccination status at the time of hospitalization. (page21, line278-286))

I disagree with the authors about the number of patients included being large. This remains a small number study, moreover since the main assessments are made between survivors and non-survivors and there were only 14 non-survivors. This limits drastically the conclusions of this study.

Thank you for your comment. We added that it is a retrospective study and has a small sample size in the limitation section as you advised.

Third limitation is that our study was the retrospective and had the small sample size. However, we tried to include variables considering various clinical situations in the analysis. (page22, line294-296)

There are some typos all along the manuscript, it should be read again and corrected. 

Thank you for your comment. We repeatedly corrected and described typos.

Reviewer 2 Report

The authors conducted a retrospective, single-center study to investigate the prognosis and clinical outcomes of nosocomial COVID-19. Of the total 14,667 patients admitted to a university teaching hospital in Korea between 11/1/2021 and 4/31/2022, 167 patients were diagnosed with nosocomial COVID-19. The authors assessed the incidence rate of nosocomial COVID-19 among their hospitalized patients during the study period and identified two independent risk factors for inpatient mortality: being unvaccinated and dyspnea. The manuscript presents clinical data about nosocomial COVID-19 from a hospital in Korea where a firm quarantine policy has been maintained since COVID-19 pandemic and adds to the body of literature regarding nosocomial COVID-19. Overall, the paper is well written and has a great interest for clinical practice.

Major comments:

1   1)   According to the related description in Study design section, SARS-CoV-2 PCR test was first preformed in patients in the hospital who showed symptoms, and then was preformed in patients who shared the hospital room with a patient or caregiver with a positive diagnosis of COVID-19. On the other hand, more than half (52%) of the nosocomial COVID-19 patients were asymptomatic. Given this high proportion of nosocomial COVID-19 patients being asymptomatic, is it possible that a certain portion of asymptomatic nosocomial COVID-19 patients, such as asymptomatic COVID-19 patients who lived in single hospital room, had missed the PCR test for COVID-19 diagnosis? If this is possible, then the incidence rate of nosocomial COVID-19 might be underestimated.

2)      Table 1 displays the data comparing nosocomial COVID-19 patients who survived versus who did not. When analyzing the impact of different diseases (“Diagnosis”), the authors treated all different disease diagnoses as a single variable with multiple categories. In my opinion, it is not appropriate. The comparison should be done for each diagnosis (disease) individually. Similarly, for analysis of variable “Department”, the comparison should be done for each department individually. Given the small sample size especially the small number of patients in one group (n=14 for non-survival in this study), a comparison of a categorical variable with too many categories would lead to very little statistical power to detect a difference.

 3)      Regarding multivariate survival analysis to identify the factors that were independently associated with inpatient mortality, the authors stated that variables with p<0.05 on univariate analysis were included in the multivariate Cox regression analysis. According to Table 3, factor “Asymptomatic state” had p-value of 0.033 (< 0.05) from its univariate analysis but it was not included in the multivariate Cox analysis, could the authors provide the reason for it? 

 4)    According to the authors, the study found a relatively high mortality rate among the nosocomial COVID-19 patients. Since the patients were hospitalized for the diagnosis of other diseases, did the authors assess the cause of the death for the patients who did not survive?  Of the 14 patients who died in the hospital, were there any patients who died of other diseases? 

 5)      In addition to the study limitations that the authors have addressed, at least there are two more: the retrospective nature of the study and the small sample size. The small sample size, especially the small number of patients who did not survive (n=14), had limited the data analyses, especially the multivariate Cox regression analysis to identify the independent risk factors for inpatient mortality. Ideally, not only the variables with p<0.05 on their univariate analysis, but also the variables that show a potential association with the outcome such as the variables with p<0.25 on their univariate analysis should be included in the initial multivariate Cox regression analysis. 

Minor comments:

1   1)     In Study design section, please replace “1 November, 2021” with “November 1, 2021”, and replace “31 April, 2022” with “April 31, 2022”.

 2)      Figure 1 (B). I assume the length of hospital stay data in each department is represented by its median value. If so, please indicate it. 

 3)      In 3.1 Clinical characteristics section, it was stated that “There was no differences in sex, age or BMI between survivors and non survivors (p>0.05 for all).”. However, Table 1 shows a significant difference in age (p=0.047) between the two groups of patients. Please correct this discrepancy.

 4)    In Figure 2 about Cox proportional survival curves, please add the starting point for the “Time (days)” on x-axis, such as “Time (days) after COVID-19 diagnosis”.

 5)       In the second paragraph of Discussion section, the sentence “This may be because surgical patients with good medical conditions were included in addition to medical patients” seems to be redundant with a similar sentence after that. Please remove it.

6) For a review convenience, I would suggest adding line numbers to the manuscript. 

Author Response

Review 2

The authors conducted a retrospective, single-center study to investigate the prognosis and clinical outcomes of nosocomial COVID-19. Of the total 14,667 patients admitted to a university teaching hospital in Korea between 11/1/2021 and 4/31/2022, 167 patients were diagnosed with nosocomial COVID-19. The authors assessed the incidence rate of nosocomial COVID-19 among their hospitalized patients during the study period and identified two independent risk factors for inpatient mortality: being unvaccinated and dyspnea. The manuscript presents clinical data about nosocomial COVID-19 from a hospital in Korea where a firm quarantine policy has been maintained since COVID-19 pandemic and adds to the body of literature regarding nosocomial COVID-19. Overall, the paper is well written and has a great interest for clinical practice.

Major comments:

1)   According to the related description in Study design section, SARS-CoV-2 PCR test was first performed in patients in the hospital who showed symptoms, and then was preformed in patients who shared the hospital room with a patient or caregiver with a positive diagnosis of COVID-19. On the other hand, more than half (52%) of the nosocomial COVID-19 patients were asymptomatic. Given this high proportion of nosocomial COVID-19 patients being asymptomatic, is it possible that a certain portion of asymptomatic nosocomial COVID-19 patients, such as asymptomatic COVID-19 patients who lived in single hospital room, had missed the PCR test for COVID-19 diagnosis? If this is possible, then the incidence rate of nosocomial COVID-19 might be underestimated.

I fully agree with your opinion. During the study period, there were 14,667 hospitalized patients, and it was confirmed that all patients were negative for COVID 19 PCR at admission. After hospitalization, Covid-19 PCR was not performed regularly in all patients, but only in cases with symptoms or close contact with a Covid-19 positive person, and additionally performed on the 4th and 7th days thereafter. Therefore, as you commented, asymptomatic nosocomial COVID-19 infections were excluded, so the actual number of nosocomial COVID-19 patients may be higher than in this study. The point you mentioned is further described in the discussion and limitation section.

In addition, it is possible that the incidence of nosocomial COVID 19 in this study was underestimated, since asymptomatic nosocomial COVID 19 patients in this study accounted for 52%. Because we screened for the occurrence of nosocomial COVID 19, focusing on symptomatic patients or contacts of COVID 19 patients, asymptomatic patients without history of contact with COVID 19 patients may not be diagnosed with nosocomial COVID 19. (page19, line235-239)

Fifth, because there were many asymptomatic nosocomial COVID 19 patients in this study, it is possible that the actual incidence of nosocomial COVID 19 infection was underestimated. To address these limitations, it is hoped that a prospective cohort study that regularly conducts COVID 19 PCR will be conducted. (page22, line299-302)

2)      Table 1 displays the data comparing nosocomial COVID-19 patients who survived versus who did not. When analyzing the impact of different diseases (“Diagnosis”), the authors treated all different disease diagnoses as a single variable with multiple categories. In my opinion, it is not appropriate. The comparison should be done for each diagnosis (disease) individually. Similarly, for analysis of variable “Department”, the comparison should be done for each department individually. Given the small sample size especially the small number of patients in one group (n=14 for non-survival in this study), a comparison of a categorical variable with too many categories would lead to very little statistical power to detect a difference.

Thank you for your advice. As you advised, the survival group and the non-survival group were compared separately for each variable in the department, diagnosis, vaccination status, and admission room. And, due to the recommendation of other reviewers, the p-value of Fisher's exact test was mainly indicated. It was rewritten with these contents in the methods and results section. The changed and added contents while rewriting Table 1 were added to the result section. In Table 1, only the department part was extracted and shown below, and all other parts were modified through statistical analysis again.

The Fisher’s exact test were used for analysis of categorical variables, and continuous variables were compared using the Mann–Whitney U test. (page7, line118-119)

 3)      Regarding multivariate survival analysis to identify the factors that were independently associated with inpatient mortality, the authors stated that variables with p<0.05 on univariate analysis were included in the multivariate Cox regression analysis. According to Table 3, factor “Asymptomatic state” had p-value of 0.033 (< 0.05) from its univariate analysis but it was not included in the multivariate Cox analysis, could the authors provide the reason for it?

Thanks for pointing this important issue. The reason for subtracting asymptomatic status was that it was a variable of large number (52%) and that asked for the same “symptom” as dyspnea, but they could never overlap. We thought it would be better not to do multivariate analysis together for variables that could never be overlapped for the same condition, such as not doing multivariate analysis for both smokers and non-smokers among the variables of smoking status. By the way, asymptomatic state or dyspnea, as recommended by another reviewer, is a very obvious variable, so it would be better to exclude it from the analysis for identifying risk factors, so symptomatic variables (both asymptomatic status and dyspnea) were excluded from the multivariate analysis.

 4)    According to the authors, the study found a relatively high mortality rate among the nosocomial COVID-19 patients. Since the patients were hospitalized for the diagnosis of other diseases, did the authors assess the cause of the death for the patients who did not survive?  Of the 14 patients who died in the hospital, were there any patients who died of other diseases?

Thanks for pointing this important issue. Our researchers reviewed together and reached a consensus that patients who died in hospital were related to nosocomial COVID-19. These details are further described in the method.

Mortality was defined that patients died during hospitalization. Records of deaths were reviewed by the authors and consensus was reached that they were related to nosocomial COVID 19. (page6, line102-104)

 5)      In addition to the study limitations that the authors have addressed, at least there are two more: the retrospective nature of the study and the small sample size. The small sample size, especially the small number of patients who did not survive (n=14), had limited the data analyses, especially the multivariate Cox regression analysis to identify the independent risk factors for inpatient mortality. Ideally, not only the variables with p<0.05 on their univariate analysis, but also the variables that show a potential association with the outcome such as the variables with p<0.25 on their univariate analysis should be included in the initial multivariate Cox regression analysis.

Thank you for your comment. We added that it is a retrospective study and has a small sample size in the limitation section as you advised.

Third limitation is that our study was the retrospective and had the small sample size. However, we tried to include variables considering various clinical situations in the analysis. (page22, line294-296)

As you advised, we analyzed the mortality risk factor with a variable with a p-value of 0.25 in the comparison between the survival group and the non-survival group. We rewrote follow analysis methods.

Factors significantly associated with survival were analyzed further in a Cox proportional hazard model to adjust for the potential confounding effects of each factor. The results are reported as the hazard ratio (HR) with 95% confidence interval (CI). The 14 variables were included sex, age, and variables in Cox’s regression analysis because all were associated with p < 0.25 in univariate analysis. (page7, line122-128)

Therefore, Cox-regression analysis was performed with variables with a p-value of less than 0.25, excluding symptomatic variables, and finally, no vaccination and chronic kidney disease were found to be variables influencing mortality.

Minor comments:

1)     In Study design section, please replace “1 November, 2021” with “November 1, 2021”, and replace “31 April, 2022” with “April 31, 2022”.

Thank you for your comment. We rewrote “November 1, 2021, and April 31, 2022.”in methods section (page4, line60)

 2)      Figure 1 (B). I assume the length of hospital stay data in each department is represented by its median value. If so, please indicate it.

Thank you for your advice. We rewrote median value in figure 1 legend and indicated median value in figure 1.

Figure 1. (A) Incidence of nosocomial COVID-19 among all hospitalized patients and of COVID-19 in the population of the country as a whole. (B) Length of median hospital stay of patients with nosocomial COVID-19 and of all patients in each department. Abbreviation: COVID-19, coronavirus disease 2019.

 3)      In 3.1 Clinical characteristics section, it was stated that “There was no differences in sex, age or BMI between survivors and non survivors (p>0.05 for all).”. However, Table 1 shows a significant difference in age (p=0.047) between the two groups of patients. Please correct this discrepancy.

I’m very sorry for our mistake. We rewrote that “Age was higher in non-survivors than in survivors (p=0.047). There were no differences in sex or BMI between survivors and non-survivors” in results. (page10, line150-151)

 4)    In Figure 2 about Cox proportional survival curves, please add the starting point for the “Time (days)” on x-axis, such as “Time (days) after COVID-19 diagnosis”.

Thanks for pointing this important issue. We redraw and rewrote figure 2, legend of figure 2, and results section.

Figure 2. Cox proportional survival curves showing the effects of no vaccination in patients with nosocomial COVID-19 (A) from the time of admission, (B) from the time of diagnosis with nosocomial COVID 19, and chronic kidney disease in patients with nosocomial COVID-19 (C) from the time of admission, (D) from the time of diagnosis with nosocomial COVID 19Abbreviation: COVID-19, coronavirus disease 2019.

5)       In the second paragraph of Discussion section, the sentence “This may be because surgical patients with good medical conditions were included in addition to medical patients” seems to be redundant with a similar sentence after that. Please remove it.

Thank you for your advice. We remove sentence of above in discussion.

6) For a review convenience, I would suggest adding line numbers to the manuscript.

Thank you for your comment. We add line numbers to the manuscript.

Round 2

Reviewer 1 Report

Thank you for taking into account all the comments. I believe the paper is now much improved.

You should harmonize the way you refer to COVID-19 along the manuscript (sometimes written COVID19, COVID 19 or COVID-19... I believe COVID-19 would be the most accurrate).

Quality of English can be improved, although all data are understandable.

Author Response

Review

We sincerely appreciate the time and effort you invested to help us improve this manuscript.

We revised the manuscript (red color) and have provided a point-by-point response to your comments below.

Thank you for taking into account all the comments. I believe the paper is now much improved.

You should harmonize the way you refer to COVID-19 along the manuscript (sometimes written COVID19, COVID 19 or COVID-19... I believe COVID-19 would be the most accurrate).

Thank you for your important advice. We rewrote “COVID-19” in manuscript

Quality of English can be improved, although all data are understandable.

Thank you for comments. We received a new manual English proofreading through MDPI's editing service.
